# LINKIN, a new transmembrane protein necessary for cell adhesion

**Mihoko Kato[1], Tsui-Fen Chou[1,2,3], Collin Z Yu[1†], John DeModena[1], Paul W Sternberg[1]***

[1]Division of Biology and Biological Engineering, Howard Hughes Medical Institute, California Institute of Technology, Pasadena, United States; [2]Division of Medical Genetics, Department of Pediatrics, Harbor-UCLA Medical Center, Torrance, United States; [3]Los Angeles Biomedical Research Institute, Torrance, United States

**Abstract** In epithelial collective migration, leader and follower cells migrate while maintaining cell–cell adhesion and tissue polarity. We have identified a conserved protein and interactors required for maintaining cell adhesion during a simple collective migration in the developing *C. elegans* male gonad. LINKIN is a previously uncharacterized, transmembrane protein conserved throughout Metazoa. We identified seven atypical FG–GAP domains in the extracellular domain, which potentially folds into a β-propeller structure resembling the α-integrin ligand-binding domain. *C. elegans* LNKN-1 localizes to the plasma membrane of all gonadal cells, with apical and lateral bias. We identified the LINKIN interactors RUVBL1, RUVBL2, and α-tubulin by using SILAC mass spectrometry on human HEK 293T cells and testing candidates for *lnkn-1*-like function in *C. elegans* male gonad. We propose that LINKIN promotes adhesion between neighboring cells through its extracellular domain and regulates microtubule dynamics through RUVBL proteins at its intracellular domain.

***For correspondence:** pws@caltech.edu

**Present address:** †School of Pharmacy, University of California, San Francisco, San Francisco, United States

**Competing interests:** The authors declare that no competing interests exist.

## Introduction

In epithelial collective migration, interconnected cells migrate together in various configurations, such as sheets, branches, chains, and clusters, to produce organs of diverse shapes and possess both epithelial and mesenchymal characteristics (*Montell, 2001*; *Haas and Gilmour, 2006*; *Ewald et al., 2008*; *Zelenka and Arpitha, 2008*). The cells develop apico-basal polarity and cell–cell adhesion as an epithelial tissue, but cells at leading edge of the group are also capable of migration. Many of the components involved in individual cell migration also affect collective cell migration (*Rorth, 2011*), such as response to external guidance cues (*Klämbt et al., 1992*; *Haas and Gilmour, 2006*; *Bianco et al., 2007*; *Pozzi and Zent, 2011*) and establishment of front–back polarity (*Prasad and Montell, 2007*; *Janssens et al., 2010*; *Ng et al., 2012*; *Law et al., 2013*; *Lebreton and Casanova, 2014*). Collective cell migration, however, additionally depends on the ability of cells to coordinate and follow the leader cells. Cell–cell adhesion molecules such as cadherins (*Cai et al., 2014*; *Menko et al., 2014*) and tissue organization through the planar cell polarity pathway (*Muñoz-Soriano et al., 2012*) impact the collective migratory ability by coordinating cytoskeleton movement. Effective collective migration therefore requires not only components promoting motility but also those that contribute to tissue integrity and coordination.

The *Caenorhabditis elegans* male gonad is shaped by a collective cell migration during larval development. It has a simple organization of one migratory leader cell, the linker cell (LC), that is followed by a stalk of adherent, passive follower cells that can be visualized in live animals (*Kimble and Hirsh, 1979*; *Kato and Sternberg, 2009*). After the migration leads the elongating gonad from its origin at the mid-body to the cloaca opening near the posterior end of the body, the gonad completes its differentiation into the mature structure. The migratory linker cell (LC) is a hybrid of mesenchymal and epithelial-like characteristics, while the follower somatic cells are epithelial-like.

**eLife digest** In animals, cells can move from one place to another to shape tissues, heal wounds, or defend against invading microbes. A cell may move alone or it may be attached to others and move as part of a group. One member of the group leads this 'collective migration', but it is not known how the cells are able to stick to each other and move together.

Collective migration takes place in the male gonad—the organ that makes sperm cells—in larvae of the nematode worm *C. elegans*. As the gonad matures, a group of cells form a simple chain that can move together. Kato et al. found that a protein called LINKIN must be present for this to happen.

LINKIN is found in the membrane that surrounds animal cells. One section of the protein—called the β-propeller—sits on the outside surface of the membrane. The structure of the β-propeller is similar to a section of another protein—called α-integrin—that also allows cells to attach, suggesting LINKIN may work in a similar way.

LINKIN is found in many animals, so Kato et al. searched for proteins that can interact with it in human cells. This search revealed three proteins that can interact with LINKIN and are required for the cells to move together. Two of the proteins control elements of the internal scaffolding of the cell: this scaffolding, which is known as the cytoskeleton, is involved in moving the cells.

The experiments suggest that LINKIN coordinates the process of binding together with the changes in the cytoskeleton that are needed to allow the cells to move as one. The next challenge is to understand how LINKIN changes the internal program of the cells to achieve this.

The cellular organization of the migrating male gonad is similar to the migrating branches in lung, trachea, and vascular development, in which interconnected cells organize into stalks behind leader tip cells (*Affolter et al., 2009*; *Eilken and Adams, 2010*). As with other branching structures (*Ikeya and Hayashi, 1999*; *Llimargas, 1999*), Notch signaling is required to specify roles between leader and follower cells in the *C. elegans* gonad (*Kimble and Hirsh, 1979*; *Greenwald et al., 1983*). However, unlike other systems, the role of the leader and follower is simplified, as they are not interchangeable once established (*Kimble, 1981*). Investigation into genes required for the migration of *C. elegans* gonadal leader cells has revealed similarities to other cell migrations, including their responding to netrin and Wnt guidance cues (*Hedgecock et al., 1990*; *Merz et al., 2001*; *Cabello et al., 2010*), binding to the extracellular matrix (ECM) through integrin receptors, and remodeling of surrounding ECM using metalloproteases (*Blelloch and Kimble, 1999*; *Nishiwaki et al., 2004*). However, little is known about the interaction between cells to promote effective collective migration.

We have identified a new protein, LINKIN, required for maintaining tissue integrity through cell adhesion and apical polarization. LINKIN is a previously uncharacterized transmembrane protein conserved among metazoans. We identified seven atypical FG–GAP domains in LINKIN that may fold into a β-propeller domain resembling the α-integrin ligand-binding domain. We show that the *C. elegans* LINKIN protein, LNKN-1, is localized to membranes of interconnected cells, most pronouncedly at apical surfaces and cell–cell contacts. In particular, LNKN-1 is required for adhesion among collectively migrating gonadal cells in *C. elegans*, although it is also expressed in many other interconnected tissues. Taking advantage of the conservation between *C. elegans* and human LINKIN, we performed SILAC based mass spectrometry on a human cell line and functional testing in *C. elegans* to identify potential interactors of LINKIN. Members of the highly conserved AAA+ ATPase family, RUVBL1 and RUVBL2, and the cytoskeletal protein α-tubulin physically interacted with LINKIN and were required for collective gonadal migration. Our data support a function for LINKIN as an adhesion molecule that uses its extracellular domain to bind molecules on the surface of neighboring cells and its intracellular domain to regulate microtubule dynamics.

## Results

### Characterizing the collective cell migration of the *C. elegans* male gonad

The developing male gonad is a collective cell migration consisting of a chain of passively migrating somatic and germ cells led by a migratory somatic cell, the linker cell (LC) (*Figure 1A–C*). After

**Figure 1**. The collective migration of the male gonad in wild-type animals and its disruption in *lnkn-1* mutants.
(**A**) Intact wild-type male gonad shape is generated by the collective migration of somatic and germ cells. The migration of the leader cell, the linker cell (LC; green), pulls the interconnected follower cells. The somatic gonad consists of the LC, the vas deferens precursors (yellow), the seminal vesicle precursors (blue), and the distal tip cells (orange). The germ cells (purple) follow behind most of the somatic gonad. At the beginning of its migration in the early L2 stage, the gonad is a small cluster of cells in the ventral mid-body of the animal (top left panel). As the LC migration defines the shape of the mature gonad, the gonad expands through the proliferation of the interconnected follower cells (bottom left panel). Longitudinal and transverse sections of the vas deferens precursor cells reveal the apical domain (red) running through the somatic gonad core (right panels). (**B–F**) Nomarski micrographs of gonads superimposed with fluorescence images of YFP-tagged LC (green fluorescence, black arrow). Gonad is outlined in the same color scheme as (**A**). (**B**, **C**) Wld-type L3 and L4 stage gonads. (**D**) The connection between the LC and the gonad was severed by ablating cells immediately behind the LC and examined 6 hr later. The LC alone has continued to migrate along its normal path, while the gonad no longer elongates after being severed
*Figure 1. Continued on next page*

*Figure 1. Continued*

from the LC. (**E**) In the L3 stage *lnkn-1(gk367)* mutant, the gonad starts to show thinning of follower cells (yellow arrows) behind the LC (black arrow). (**F**) By the mid-L4 stage, gonad (yellow arrows) has stopped migrating at the point where the LC detached, while the LC (black arrow) has continued to migrate. In this and subsequent figures, anterior (A) is to the left, posterior (P) is to the right, dorsal (D) is to the top, and ventral (V) is to the bottom.

migration, the interconnected somatic cells behind the LC differentiate during the transition from the fourth larval (L4) stage to the adult into a mature gonad structure, a tube comprising the vas deferens and seminal vesicle. Behind the somatic gonad are the proliferating germ cells, arranged from the newest in the distal region to the most developed closest to the somatic gonad. Capping the distal end of the gonad are the two male distal tip cells, which maintain the mitotic germ cells. To form this gonad shape during the L2 through L4 stages of the larval development, the LC leads the elongating gonad from the mid-body region to the cloaca opening in the posterior body, where it dies after completing the migration. During the L3 stage of the migration, the somatic cells of the vas deferens and seminal vesicle precursors divide from seven to 53 cells to form the elongating gonad. The developing somatic gonad has epithelial-like characteristics consisting of strong intercellular connections and a developing apical domain running down the core of the gonad (*Figure 1A*). The somatic cells have a radial symmetry around this core, and as they proliferate, the daughter cells are incorporated into the chain while maintaining this configuration.

We examined the requirement of different gonadal cell types for migration. By ablating the LC (15/15), we confirmed the findings of *Kimble (1981)* that the LC is necessary for migration and is not replaced by a follower cell taking on its migratory role. Without the LC, the gonad stopped elongating but continued to balloon through cell proliferation. The LC, however, was capable of migrating alone if the somatic cells around it were ablated in the L3 stage (8/10). In this case, the LC alone migrated along its normal course while the gonad stopped elongating at the point of LC detachment (*Figure 1D*). Since LC migration was slower than normal, the LC often did not complete its migration by the L4-to-adult transition. The cause of the slower migration may be due to the missing contribution of gonadal cells or other factors such as drag from scarred tissue. The germ cells add to gonadal mass but are not necessary for migration, as the LC reaches the cloaca even when germ cell precursors are ablated.

## LNKN-1 is required for gonadal cells to migrate collectively

We discovered the *lnkn-1* mutant during a process of identifying new genes involved in LC migration by utilizing a database of expression patterns reported by the Genome BC *C. elegans* Gene Expression Consortium (*Hunt Newberry et al., 2007*). Since the consortium only reports hermaphrodite expression patterns, we searched their database for genes expressed in the migratory leader cells for the hermaphrodite gonad, the distal tip cells (DTCs), which functionally are the closest cells to the male LC (*Kimble and Hirsh, 1979*). We reasoned that the gonadal leader cells of both sexes may use partially overlapping genes for their migrations. One of the genes reported to be expressed in the hermaphrodite DTCs was *ZK637.3* (WBGene00014023), a conserved gene of unknown function that had an available deletion mutant, *gk367*. After obtaining this mutant and observing unusual male gonad defects, we decided to investigate this gene further. We have renamed this gene from *tag-256* (*temporarily assigned gene-256*, ZK637.3) to *lnkn-1* (*LiNKiNg-1*).

In males homozygous for *lnkn-1(gk367)*, gonadal cells near the LC became detached during gonad migration (n = 29/30) such that the LC continued to migrate, either alone or with a few remaining follower cells, but the rest of the gonad did not follow. This detachment resulted in a partially elongated gonad and, further ahead along the normal path, a detached LC alone or with a few adherent follower cells (*Figure 1E,F*). This phenotype is similar to that of the gonad with ablated follower cells behind the LC (*Figure 1D*). The position of detachment was variable but usually occurred within a few cell lengths behind the LC, suggesting that the pulling force generated by the LC may have caused detachment. Although the LC continued to migrate along its normal course after detachment and occasionally completed the migration, the male was sterile since the gonad did not connect to the cloaca opening. In *lnkn-1(gk367)* mutant hermaphrodites, the gonadal leader DTCs remained connected but migrated a shorter distance than the wild type and their shape appeared elongated

and strained. The mutant also recessively caused maternal effect lethality (n = 30/30). The male gonad phenotype in *lnkn-1(gk367)* mutants suggested that *lnkn-1* is required for cell–cell interaction rather than LC migration.

## LINKIN is a conserved transmembrane protein

LNKN-1 was a conserved, poorly characterized protein predicted to be a type I single-pass transmembrane protein of 599 amino acids (AA), consisting of a 19 AA signal sequence, 533 AA extracellular domain, 23 AA transmembrane domain, and 24 AA intracellular domain (*Figure 2*). We were able to identify homologs of LNKN-1 back to an early branching animal phylum, Placozoa, as well as in fungi, and have called this protein family LINKIN. The presence of LINKIN in *Plasmodium falciparum* has previously been noted (*Kaczanowski and Zielenkiewicz, 2003*). LINKIN is conserved in Metazoa from Placozoa *Trichoplax adhaerens*, a basal metazoan, to vertebrates including human. A protein alignment of LINKIN from *Homo sapiens* (ITFG1/TIP, 612 AA), *Mus musculus* (ITFG1/TIP, 610 AA), *Drosophila melanogaster* (CG7739, 596 AA), and *C. elegans* (LNKN-1, 599 AA) revealed that all orthologs have similar protein lengths and domain organizations (*Figure 2B*).

Overall, the protein sequence between *H. sapiens* and *C. elegans* excluding the signal sequence is 26% identical (154 AA) and 61% similar (365 AA). However, LINKIN has a highly conserved intracellular domain, which is 62.5% (15/24 AA) identical and 87.5% similar (21/24 AA, clustalo analysis). In particular, the last eight amino acids are identical in all four species (*Figure 2B*). Despite its high conservation, a BLAST search of the intracellular domain alone did not identify strong similarities with domains in other proteins.

Protein motifs in LINKIN were largely unknown, with the only ascribed motif being an FG–GAP domain found in one copy in *H. sapiens* and three in *M. musculus* (uniprot.org). The FG–GAP is a domain that occurs in seven copies in α-integrins and folds into a seven-bladed β-propeller structure that serves as its ligand-binding domain (*Springer, 1997*; *Xiong et al., 2002*). We have identified seven atypical FG–GAP domains in the N-terminal of the extracellular domain, based on sequence similarities to the annotated LINKIN FG–GAP domains from *H. sapiens* and *M. musculus* and to α-integrin FG–GAP domains (*Figure 2*, 'Materials and methods'). FG–GAP domains have a loosely conserved Phe-Gly and Gly-Ala-Pro sequence, which are separated by sequence that can include a calcium-binding motif. A comparison of all human α-integrin FG–GAP domains showed that their calcium-binding motif has a strong DxxxDxxxD signature (D = Asp, x = AA; *Chouhan et al., 2011*). We found a strong DxxxDxxxD signature in all seven calcium-binding domains of LINKIN, suggesting that these are similar to FG–GAP domains of α-integrins (*Figure 2*). The significance of finding seven FG–GAP domains in the N-terminal of LINKIN is the possibility that LINKIN, like α-integrins, uses a seven-bladed β-propeller structure to bind ligand. We also identified a highly conserved extracellular region adjacent to the transmembrane domain, which is a yet unrecognized protein domain (*Figure 2*). LINKIN has a dozen predicted N-linked glycosylation sites (uniprot.org). Taken together, our investigation shows that LINKIN is a conserved transmembrane glycoprotein pre-dating Metazoa and potentially containing a seven-bladed, β-propeller, ligand-binding domain.

## LNKN-1 is expressed in the apical and lateral membrane of tissues

We examined the expression pattern and subcellular localization of LNKN-1 in *C. elegans*, particularly in the male gonad. Previously, *lnkn-1* localization was categorized to be in cell membrane, when examined by GFP-tagging in a screen of muscle-related genes (*Meissner et al., 2011*). The only other investigation into LINKIN observed that in mammals the extracellular domain functions as a secreted protein that modulates T-cell dependent immune response (*Fiscella et al., 2003*). We made both extracellularly and intracellularly YFP-tagged versions of LNKN-1 expressed under its natural regulatory specific promoter (*Figure 3E,F*, *Figure 3—figure supplement 2M,N*). Both YFP::LNKN-1 and LNKN-1::YFP are similarly localized to the plasma membrane of many cells. LNKN-1 begins to be expressed in all somatic gonadal cells of the male, including the LC, the vas deferens precursor cells, and seminal vesicle precursor cells, starting in the early L3 stage and continuing through adulthood (*Figure 3E,F*). It is also expressed in all somatic gonadal cells of the hermaphrodite, including the distal tip cells, anchor cell, uterine precursor cells, and spermatheca precursor cells (*Figure 3—figure supplement 2*). Other expression occurs in pharynx, pharyngeal-intestinal valve, intestine, excretory cell and canal, seam cells, a specialized subset of hypodermal cells, the vulval precursor cells of the hermaphrodite, and hook precursor cells in the male (*Figure 3—figure supplement 2*). YFP-tagged

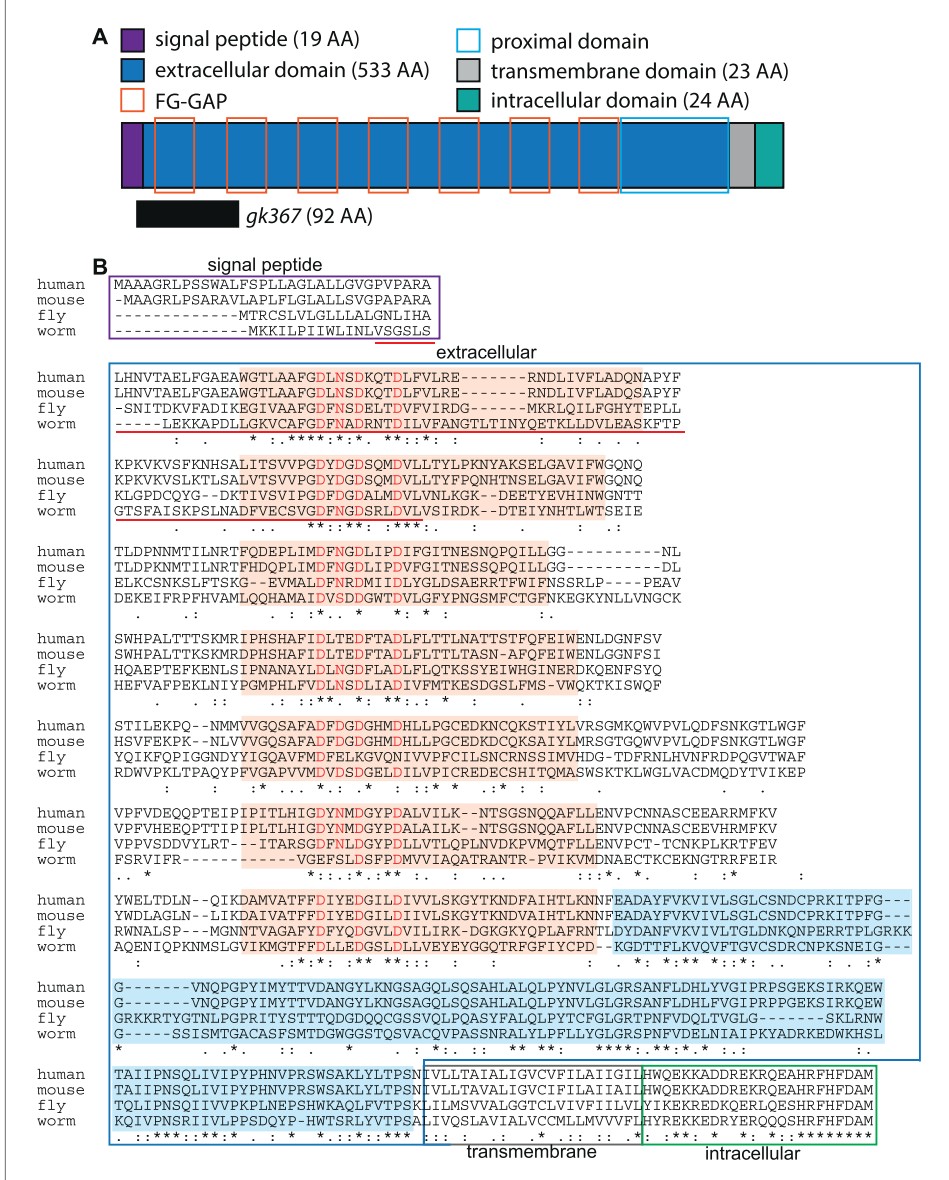

**Figure 2**. LINKIN protein domains and sequence are conserved across diverse metazoan species. (**A**) *C. elegans* LNKN-1 is a single-pass transmembrane protein of 599 amino acids (AA). Conserved protein motifs include seven atypical FG–GAP domains (orange boxes) and an extracellular region proximal to the transmembrane domain (light blue box). The *gk367* genomic lesion results in the deletion of 92 AAs based on cDNA sequencing. (**B**) LINKIN sequence from divergent animals (*Homo sapiens*, *Mus musculus*, *Drosophila melanogaster*, and *Caenorhabditis elegans*) were aligned using Clustalw. The intracellular domain shows high conservation in all examined species. '*' indicates identical AA, ':' indicates strong similarity, and '.' indicates weak similarity. Signal peptide is boxed in purple, extracellular domain in blue, transmembrane domain in gray, and intracellular domain in green. Sequence deleted by *gk367* mutation is underlined in red. FG–GAP domains are highlighted in orange, and the Dx(D/N) xDxxxD calcium-binding motif contained within each FG–GAP domain is indicated with red letters. FG–GAP domains were defined as a region from 8 AA N-terminal of the calcium-binding domain to 18 AA C-terminal of the calcium-binding domain, based on annotation by uniprot.org of the second, third, and fifth FG–GAP domains in *M. musculus* LINKIN.

LNKN-1 is localized to the plasma membrane, exhibiting stronger localization to the sides of cell–cell contact in tissues such as the intestine, seam, and gonad. This broad expression is consistent with our observations that *lnkn-1* is expressed in the LC but not enriched (*Schwarz et al., 2012*).

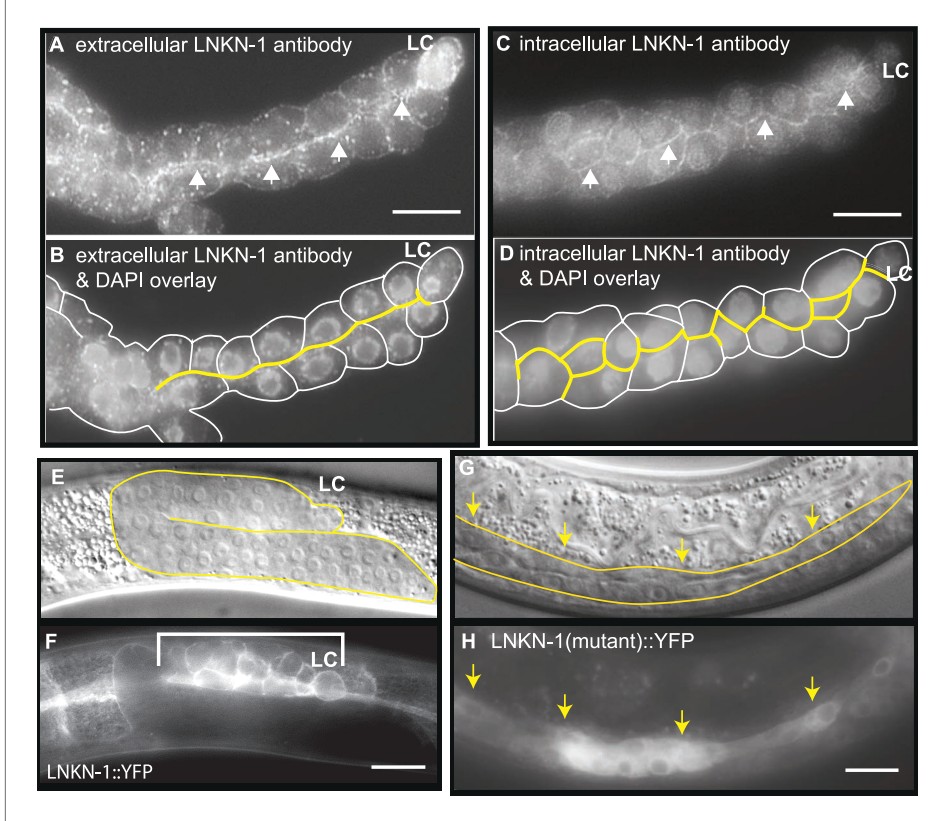

**Figure 3**. LNKN-1 localizes to the apical and lateral plasma membrane. (**A–D**) Immunofluorescence staining of dissected male gonads using antibodies against LNKN-1 extracellular domain (**A**) and intracellular domain (**C**) shows localization to the plasma membrane with enrichment at lateral and apical regions (arrows). The antibody against the extracellular domain also labeled cytoplasmic puncta (**A**), and the antibody against the intracellular domain also labeled the nucleus (**C**), but these may be due to non-specific staining since they were present in gonads from *lnkn-1* RNAi-treated and mutant animals (***Figure 3—figure supplement 1***). (**B**) An overlay of the image from (**A**) and an image of the same gonad stained with DAPI. (**D**) An overlay of the image from (**C**) and an image of the same gonad stained with DAPI. In both (**B**) and (**D**), the cells are outlined in white and the apical domain is highlighted in yellow. (**E** and **F**) Nomarski and epifluorescence images of a live animal expressing YFP-tagged LNKN-1 show that LNKN-1::YFP is expressed in the plasma membrane of gonadal cells but has spread to the basolateral domain. Bracket marks the male somatic gonad. (**G** and **H**) LNKN-1(mutant)::YFP, in which wild-type *lnkn-1* cDNA construct from (**F**) is replaced by *lnkn-1*(*gk367)* mutant cDNA, does not localize to the plasma membrane. Nomarski (**G**) and epifluorescence (**H**) images are of male somatic gonad from a live animal expressing LNKN-1(mutant)::YFP. Scale bar represents 10 µm. Anterior is to the left, posterior is to the right, dorsal is to the top, and ventral is to the bottom.

The following figure supplements are available for figure 3:

**Figure supplement 1**. LNKN-1 antibodies specifically label LNKN-1 protein in the plasma membrane.

**Figure supplement 2**. Expression pattern for YFP-tagged LNKN-1.

**Figure supplement 3**. *lnkn-1* RNAi silencing reduces LNKN-1 protein and causes gonad cell detachment defects.

To examine the localization of the native protein, two polyclonal antibodies were raised in rabbit against the entire extracellular domain (533 AA) of LNKN-1 and against a peptide derived from the last 17 AA of the short intracellular domain and affinity-purified with the respective antigens. Immunofluorescence staining was performed on dissected gonads and intestines, which greatly improves antibody penetration over whole animals (***Figure 3A–D***). Staining of other dissected parts of the worm confirmed expression in the pharynx, excretory canal, and seam cells, indicating that the

tissue expression pattern of LNKN-1::YFP is accurate. However, there was an important difference in subcellular localization between the native and YFP-tagged proteins: the antibodies showed stronger localization of LNKN-1 to the apical and lateral domain of the gonad (*Figure 3A,B*) and intestine, while YFP-tagged LNKN-1 is uniformly distributed in the plasma membrane. While both antibodies show heavier localization to the apical and lateral plasma membrane, the extracellular domain antibody shows additional staining in large cytoplasmic puncta (*Figure 3A*), and the intracellular domain antibody shows nuclear staining (*Figure 3B*, *Figure 3—figure supplement 1G*). In *lnkn-1* mutants and *lnkn-1* RNAi-treated animals, we do not see plasma membrane staining with antibodies against either the extracellular or intracellular domain, but we do see non-specific staining in the cytoplasm and nucleus (*Figure 3—figure supplement 1*). This indicates that the membrane staining is due to LNKN-1 localization, but the cytoplasmic and nuclear stainings are due to non-specific binding of other proteins. The antibodies also stain the plasma membrane in the germ cells, where YFP expression could not be determined because germ cells do not readily express transgenes. The two antibodies reveal the true localization of native LNKN-1 to be in the plasma membrane with a preference for apical and lateral domains.

### Characterization of the *lnkn-1(gk367)* deletion

Since this was the first reported use of the *lnkn-1(gk367)* deletion allele, we characterized it molecularly. The genomic locus of *lnkn-1* spans 3640 nucleotides and is the second gene in an operon. The *gk367* deletion of *lnkn-1* excises 393 base pairs (bp) of genomic DNA starting soon after the signal sequence and ending in an intron. Since a possibility existed that an in-frame mRNA could be transcribed from the deletion, we characterized the truncated mRNA through RT-PCR and sequencing. The *lnkn-1* cDNA sequence resulting from the *gk367* deletion is missing the last 18 bp of the signal sequence and the first 258 bp of the extracellular domain (*Figure 2*). The larger size of the cDNA deletion than would be predicted based on the genomic lesion indicates that an alternate splice site was used when the lesion removed the usual splice donor; however, the product cDNA is still in-frame.

Since mRNA was being transcribed in *lnkn-1* mutants, we investigated whether protein was being expressed. We generated a *lnkn-1 promoter::lnkn-1(mutant cDNA)::yfp* construct, in which yfp was fused to mutant *lnkn-1* cDNA and expressed using its 5′ genomic region. We found that LNKN-1(mutant)::YFP is in fact expressed but shows cytoplasmic rather than plasma membrane expression (*Figure 3G,H*). This confirms that a truncated protein is being produced from the *lnkn-1* deletion locus but is mislocalized and unlikely to have its normal function.

### RNAi of *lnkn-1* supports *lnkn-1(gk367)* functioning as a null mutant

To test whether the *lnkn-1* deletion mutant functions as a null despite producing a truncated protein, we examined the phenotype produced by *lnkn-1* RNAi silencing. Males treated with *lnkn-1* RNAi produced gonadal defects that were milder than the mutant (*Figure 3—figure supplement 3A*). While only 11% (4/35 animals) had detached gonadal cells, an additional 17% (6/35 animals) had 'stringy' gonads, in which fewer cells remained attached and were stretched from pulling by the LC. We also performed *lnkn-1* RNAi on animals expressing LNKN-1::YFP to ensure that the RNAi was effective (*Figure 3—figure supplement 3B–E*). LNKN-1::YFP was absent from the gonad in all animals (n = 28), but was retained in a few tissues including the pharynx and excretory canal, likely because these tissues produce higher levels of LNKN-1 or were more resistant to the effects of RNAi. Since RNAi effectively reduces but does not eliminate the function of LNKN-1, we interpret the similar but stronger phenotype of the mutant to indicate that the mutant is in fact a loss-of-function allele.

### Full-length LNKN-1 is required to rescue the mutant phenotype

We investigated the requirement of various domains of LNKN-1 for rescuing the mutant phenotype. We were able to completely rescue the *lnkn-1* mutant, including maternal effect lethality and gonad adhesion defects, using a genomic construct (*Figure 4A*). Since *lnkn-1* is the second gene in an operon, this construct contains 4.5 kb of genomic region upstream of *lnkn-1* start site, the *lnkn-1* gene, and the *lnkn-1* 3′ UTR (*Figure 4A*). We also made a cDNA construct using the same 5′ region of *lnkn-1* fused to *lnkn-1* cDNA and *unc-54* 3′ UTR, a common 3′ UTR for *C. elegans* constructs. The *lnkn-1* cDNA was able to rescue the gonad defect but not maternal effect lethality (*Figure 4B*), possibly because it does not contain all regulatory elements for complete tissue expression or is silenced in the germline.

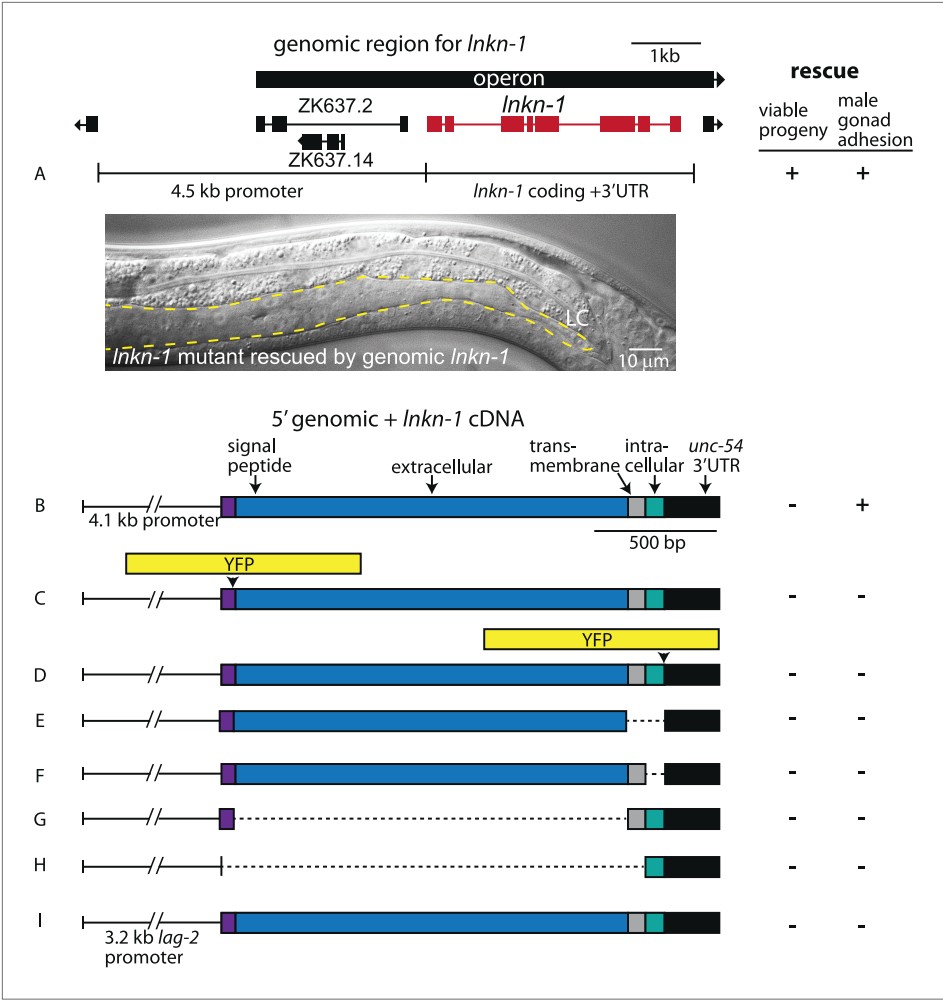

**Figure 4**. Rescue of *lnkn-1* mutant phenotypes requires the full-length *lnkn-1* gene. (**A**) A full-length genomic construct containing 4.5 kb of 5' regulatory region, *lnkn-1* coding region, and 3' UTR rescues both gonad detachment defects in the male and maternal lethality. '+' indicates rescue and '−' indicates no rescue. Micrograph shows the posterior body of a male *lnkn-1(gk367)* mutant that has been rescued for gonad detachment by a full-length genomic *lnkn-1* construct. (**B**) 4.1 kb of genomic promoter region fused to *lnkn-1* cDNA and *unc-54* 3' UTR rescues male gonad defect but not maternal lethality. (**C** and **D**) Constructs with YFP inserted within the extracellular domain (**C**) or the intracellular domain (**D**) of *lnkn-1* cDNA did not rescue *lnkn-1* mutants. (**E–H**) 4.1 kb of genomic 5' control region fused to partial domains of *lnkn-1*(cDNA) do not rescue. *lnkn-1* constructs of extracellular domain only (**E**), extracellular and transmembrane domain (**F**), transmembrane and intracellular domain (**G**), and intracellular only (**H**) also did not rescue *lnkn-1* mutants. (**I**) LC-specific expression of *lnkn-1* using *lag-2* control sequences also does not rescue.

We also investigated whether other *lnkn-1* constructs can rescue the mutant (***Figure 4***). Full-length *lnkn-1* with *yfp* fused to either the intracellular or extracellular domain did not rescue the mutant (***Figure 4C,D***). This is not surprising considering that YFP tagging prevents correct localization of LNKN-1 to the apical domain (see above). While both YFP-tagged proteins were expressed well in wild-type animals, intracellularly fused LNKN-1::YFP is not well-tolerated in *lnkn-1* mutant heterozygotes and is lost within a few generations.

Partial domains of *lnkn-1* were also not able to rescue the mutant; we attempted rescue with constructs containing only the secreted extracellular domain, the extracellular and transmembrane domain, the transmembrane and intracellular domain, and the cytoplasmic intracellular domain (***Figure 4E–H***). Lastly, we expressed full-length *lnkn-1* under a LC promoter to test a gonadal non-cell autonomous effect, but this also did not rescue the mutant (***Figure 4I***). We conclude that the intact protein, with intact extracellular and intracellular domains, is required for the function of LNKN-1.

## Identifying interactors of human LINKIN

Since no interactors were previously known, we used a proteomics approach to identify binding partners in order to better understand the function of LNKN-1 through its interactions. Based on the hypothesis that the highly conserved intracellular sequence suggests both a required function for this domain and a potential for its binding partners to be conserved, we decided to identify LINKIN interactors by mass spectrometry using a human cell line. The advantage of using a human cell line compared to whole *C. elegans* is that it is a homogeneous cell type and the conditions for SILAC mass spectrometry are established. By using SILAC (stable isotope labeling by amino acids in cell culture), experiment and control immunoprecipitates can simultaneously be analyzed by mass spectrometry, enabling both quantitation and background subtraction. SILAC mass spectrometry was performed on immunoprecipitates from human LINKIN-Myc-expressing HEK 293T cells without isotopic labeling and from non-transfected cells with heavy isotopic labeling. Based on the ratio of 'light' to 'heavy' isotopes for each protein, enrichment through specific binding to LINKIN over background non-specific binding was determined. 484 proteins with at least two unique peptides were identified by LC/MS/MS from the immunoprecipitate with LINKIN, excluding common contaminants (*Supplementary file 1*). As validation for successful immunoprecipitation, LINKIN was itself one of the proteins with highest enrichment in LINKIN-Myc-expressing cells over control cells (35-fold enrichment; *Figure 5A*).

## RUVB-1, RUVB-2, and α-tubulin function with LNKN-1 in gonad cell adhesion

Our aim was to identify, among the many LINKIN interactors, those functioning in maintaining gonadal cell attachment. We performed an assay in *C. elegans* based on the hypothesis that some of the binding partners of human LINKIN would also be conserved in *C. elegans*. Our approach was to assign *C. elegans* homologs to the ITFG1-interacting genes and perform an RNAi screen in *C. elegans*, seeking genes with a similar gonadal cell detachment phenotype as the *lnkn-1* mutant. There were 68 proteins (excluding LINKIN) identified by mass spectrometry that were >fivefold enriched in ITFG1-Myc immunoprecipitates over control, and 45 of them had at least one *C. elegans* homolog (*Supplementary file 2*). 40 genes were available in existing RNAi libraries and screened. The silencing of three genes, *ruvb-1*/RUVBL1 (also known as Pontin), *ruvb-2*/RUVBL2 (also known as Reptin), and *tba-2*/α-tubulin, caused a similar gonadal defect to *lnkn-1*, (*Figure 5B–E*). *ruvb-1*/RUVBL1 and *ruvb-2*/RUVBL2 are highly conserved members of the AAA+ ATPase superfamily of proteins that often function together in a hexameric ring complex (*Matias et al., 2006*; *Gorynia et al., 2011*). α-tubulin together with β-tubulin forms microtubules, which as part of the cell cytoskeleton have roles in cell mechanics and transport of cellular components (*Etienne-Manneville, 2013*). β-tubulin was also among the highly enriched human gene interactors, but the homologous *C. elegans* β-tubulin did not produce a gonadal phenotype by RNAi. We tested the other five *C. elegans* β-tubulin genes and found that *tbb-2* has a gonadal detachment defect (*Figure 5F*).

## LINKIN binds RUVBL1, RUVBL2, and α-tubulin at the plasma membrane

Having identified RUVBL1, RUVBL2, α- and β-tubulin as potential interactors of LINKIN that were also involved in the same biological process as LNKN-1 in cell adhesion, we wanted to confirm their physical interaction. Binding between RUVBL1 and RUVBL2 into heteromeric multimers (*Gorynia et al., 2011*) and binding between RUVBLs and microtubules have been reported (*Gartner et al., 2003*; *Dobreva et al., 2008*; *Ducat et al., 2008*). To test physical interaction between LINKIN and each of RUVBL1, RUVBL2, α- and β-tubulin, we performed Western blots on co-immunoprecipitates from ITFG1-Myc-expressing HEK 293T cells. Probing with antibodies against RUVBL1, RUVBL2, α-tubulin, and β-tubulin, we found that RUVBL1, RUVBL2, and α-tubulin bound LINKIN (*Figure 5G–I*). However, we did not observe β-tubulin binding LINKIN (*Figure 5J*). As a control for binding specificity of abundant cytoskeletal proteins, we also probed with an anti-β-actin antibody and found that β-actin did not bind LINKIN (*Figure 5K*).

We next investigated whether these interactions occur at the plasma membrane. We isolated the membrane and cytoplasmic fractions from HEK293T cells transfected with plasmids expressing ITFG1-Myc, Flag-RUVBL1, and HA-RUVBL2. By Western blot analysis, we showed that RUVBL1, RUVBL2, and α-tubulin were present in both cytoplasmic and membrane fractions, but ITFG1 was only present in the membrane fraction. We then immunoprecipitated with Flag-RUVBL1 from both cytoplasmic and membrane fractions and determined by Western blot analysis which proteins interacted with

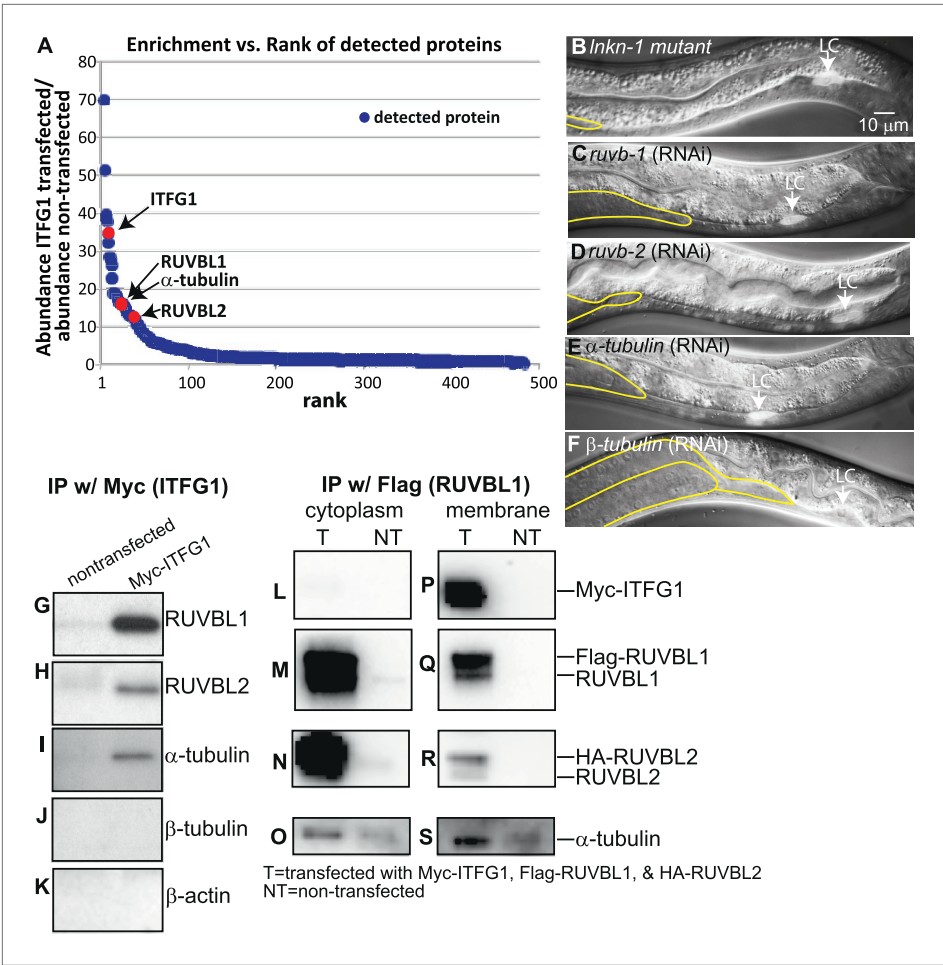

**Figure 5**. Interactors of LINKIN are RUVBL1, RUVBL2, and α-tubulin. (**A**) Graph represents human LINKIN interactors identified by mass spectrometry. ITFG1 (human LINKIN) co-immunoprecipitates from SILAC treated HEK 293T cells with ITFG1-Myc expression were compared to unlabeled cells without ITFG1-Myc expression. (**B–F**) RNAi knockdowns in *C. elegans* of *ruvb-1*, *ruvb-2*, α-tubulin, and β-tubulin show the same gonad cell detachment as *lnkn-1* mutant. Gonad is outlined in yellow and LC is marked by cytoplasmic YFP. Figures are an overlay of Nomarski and fluorescence images. (**G–K**) Western blots of ITFG1 (human LINKIN) co-immunoprecipitates probed with antibodies against RUVBL1, RUVBL2, α-tubulin, β-tubulin, and control β-actin show that LINKIN interacts with RUVBL1, RUVBL2, and α-tubulin. Myc immunoprecipitation was performed on non-transfected cells (left column) and ITFG1-Myc transfected cells (right column). Equal protein loading was determined by Ponceau S staining. (**L–S**) Lysates from cells transfected with Myc-ITFG1, Flag-RUVBL1, and HA-RUVBL2 (labeled 'T') or with Myc control (labeled 'C') were separated into a cytoplasmic and a membrane fraction. The cytoplasmic fraction (**L–O**) and membrane fraction (**P–S**) were immunoprecipitated using Flag-RUVBL1 and assayed by Western blot for interactors. ITFG1 is only detected in the membrane fraction (**P**). RUVBL1 (**M**), RUVBL2 (**N**), and α-tubulin (**O**) interact in the cytoplasmic fraction even without ITFG1. ITFG1 (**P**), RUVBL1 (**Q**), RUVBL2 (**R**), and α -tubulin (**S**) interact in the membrane fraction.

RUVBL1 (*Figure 5L–S*). ITFG1, RUVBL2, and α-tubulin interacted with RUBVL1 in the membrane fraction (*Figure 5P–S*). In the cytoplasmic fraction, RUVBL1, RUVBL2 and α-tubulin interact even in the absence of ITFG1 (*Figure 5L–O*). These results showed that the interaction between ITFG1, RUVBL1, RUVBL2, and α-tubulin occurs at the membrane, likely the plasma membrane based on *C. elegans* LINKIN localization.

## Localization of RUVB-1, RUVB-2, and α-tubulin in *C. elegans* gonad

We next investigated whether LNKN-1 interactors also localize to the plasma membrane. To examine RUVB-1 and RUVB-2 localization in the gonad, we generated polyclonal rabbit antibodies against

full-length *C. elegans* RUVB-1 and RUVB-2 proteins, which were affinity-purified using the respective full-length proteins (*Figure 6A–B*). Strong staining for both RUVB-1 and RUVB-2 was observed in the cytoplasm and nucleus of gonadal cells (*Figure 6A,B*). We demonstrated the specificity of our anti-RUVB-1 and anti-RUVB-2 antibodies by comparing their staining in dissected gonads from wild-type animals (n = 6) and *ruvb-1* (n = 6) or *ruvb-2* (n = 8) RNAi-treated animals (*Figure 6—figure supplement 1*). We observed a consistent decrease in staining in the cytoplasm and nucleus of the gonads from RNAi-treated animals, indicating that RUVB-1 and RUVB-2 actually localize to both locations. Anti-α-tubulin antibody stained a dense network of microtubule fibers throughout the cytoplasm but particularly in the cell cortex of all gonadal cells (*Figure 6C*). The microtubules were more densely aligned along the developing apical domain of the gonad. Based on the Western blot analysis of membrane fractionated cells described above and the antibody staining results in *C. elegans*, the localization of LINKIN, RUVBL1, RUVBL2, and α-tubulin intersect at the cytoplasmic face of the plasma membrane, which agrees with the adhesion function of LINKIN.

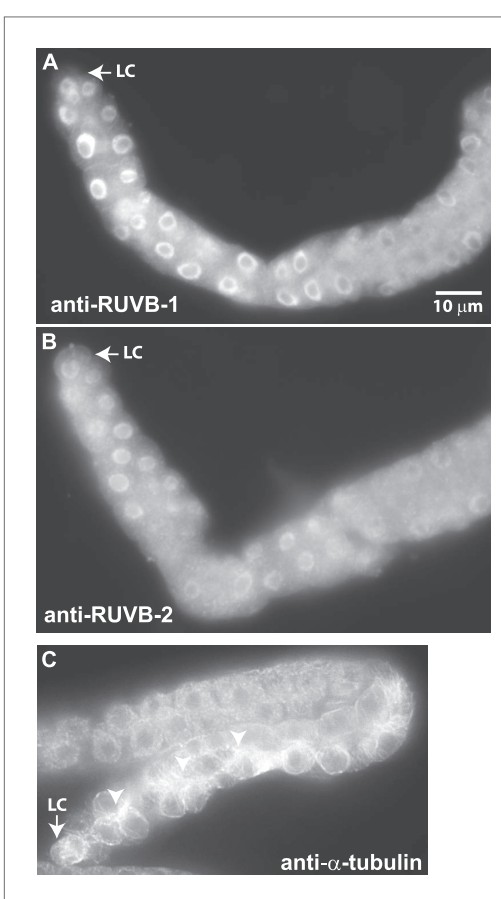

**Figure 6**. Antibodies against RUVB-1, RUVB-2, and α-tubulin show localization in *C. elegans* male gonads. Dissected male gonads stained with antibody against *C. elegans* RUVB-1 (**A**) and RUVB-2 (**B**) show localization in cytoplasm and nucleus. *Figure 6—figure supplement 1* shows that both cytoplasmic and nuclear stainings are specific to RUVB-1 or RUVB-2 proteins and can be reduced by *ruvb-1* or *ruvb-2* RNAi. (**C**) Dissected male gonad stained with antibody against α-tubulin shows network of microtubule fibers throughout the gonad, with stronger localization to the cell cortex and apical domain (arrow).

The following figure supplement is available for figure 6:

**Figure supplement 1**. The anti-RUVB-1 antibody specifically labels RUVB-1 protein.

## LNKN-1 expression and plasma membrane localization do not depend on RUVB-1, RUVB-2, and α-tubulin function

Since the known functions for RUVB-1 and RUVB-2 include transcriptional regulation (*Jha and Dutta, 2009*) and for α-tubulin include protein transport (*Miller et al., 2009*), we investigated whether RUVB-1, RUVB-2, or α-tubulin is required for either LNKN-1 expression or localization. We used RNAi to reduce *ruvb-1*, *ruvb-2*, and *tba-2* function in LNKN-1::YFP animals and found that RNAi silencing of these genes did not affect LNKN-1::YFP function or localization (*Figure 7*). For *ruvb-1* and *ruvb-2* RNAi-treated animals, we also examined LNKN-1 expression by immunofluorescence staining and found no difference from untreated animals.

## LNKN-1 remains at the plasma membrane during mitosis

RUVBLs are members of many complexes (*Rosenbaum et al., 2013*), but their only known interaction with microtubules has been at the mitotic spindle (*Gartner et al., 2003*; *Ducat et al., 2008*). We therefore investigated whether LINKIN or cleaved domains of LINKIN may also localize to the spindle during cell division. There are precedents for membrane-associated proteins involved in cell adhesion, such as ILK and β-catenin, to localize to the spindle with RUVBLs and microtubules (*Kaplan et al., 2004*; *Fielding, et al., 2008*; *Dobreva et al., 2008*). We first examined microtubule localization, which is known to redistribute during mitosis to a formation that radiates out from the mitotic spindle and attaches to the kinetochore and cell cortex. In fixed dissected gonads from L3 stage animals, by using an antibody

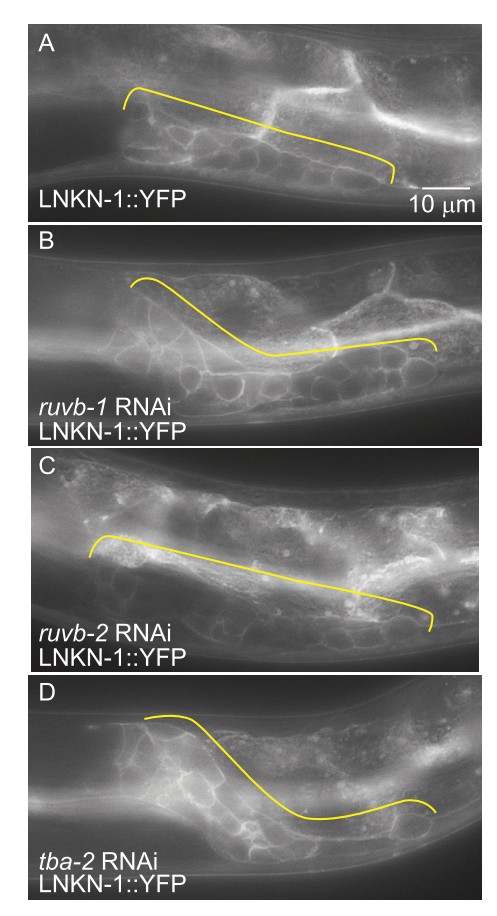

A

LNKN-1::YFP 10 µm

B

*ruvb-1* RNAi
LNKN-1::YFP

C

*ruvb-2* RNAi
LNKN-1::YFP

D

*tba-2* RNAi
LNKN-1::YFP

**Figure 7**. LNKN-1 expression and localization are not dependent on *ruvb-1*, *ruvb-2*, or *tba-2* function. Representative epifluorescence images of LINKIN::YFP animals that were either not treated (**A**), or treated with RNAi against *ruvb-1* (**B**), *ruvb-2* (**C**), or *tba-2* (**D**). The expression of LINKIN::YFP and its localization to the plasma membrane is similar in RNAi-treated and untreated animals.

against α-tubulin to identify microtubules and DAPI staining to identify condensed chromosomes of dividing cells, we observed the expected microtubule redistribution to the mitotic spindle (*Figure 8A,B*). The usually dense network of microtubules found in the cytosol and cell periphery of non-mitotic cells was replaced in mitotic cells by radial microtubules. We next examined whether LNKN-1 or domains of LNKN-1 were redistributed during cell division. Both antibodies against LNKN-1 extracellular (*Figure 8C,D*) and intracellular domains (*Figure 8E,F*) showed that localization of LNKN-1 remains unchanged at the plasma membrane during mitosis. These observations imply that there is a temporary loss of interaction between α-tubulin, RUVBL proteins, and LNKN-1 at the membrane and the interactions must be reestablished after cell division.

## LNKN-1 functions in cell–cell attachment in the gonad before mature junctions form

To determine whether gonad detachment in *lnkn-1* mutants involves the dissolution of mature cell–cell junctions, we examined cell junction formation using the apical junction marker, AJM-1::GFP, and gap junction marker innexin, INX-5::GFP (*Figure 9*). AJM-1 localizes to apical adhesion junctions in a complex with cadherins (*Köppen et al., 2001*). In the tube-shaped adult gonad, we observed AJM-1::GFP lining the entire lumen composed of the apical surfaces of the surrounding somatic gonad cells (*Figure 9C,D*). This apical accumulation begins in the mid-L4 stage as puncta (*Figure 9A,B*), but it is not present in the L3 stage when gonad detachment usually occurs in the *lnkn-1* mutant. INX-5 shows expression and localization to cell–cell junctions in a few cells in the distal somatic gonad also starting in the L4 stage and becoming stronger in the adult (*Figure 9E–H*). Results based on both adhesion markers are consistent with the timeline of gonadal development, which dictates that cell division and rearrangement occurs during the L3 stage, while differentiation into the mature structure occurs during the latter half of the L4 stage up until the transition into adulthood. Cell dissociation in *lnkn-1* mutants occurs at a stage when cell junctions are being rearranged; during this growth phase, adhesion-promoting genes like *lnkn-1* may have a greater effect than later after the establishment of other more secure junctions.

## Discussion

We have identified a new family of conserved protein, LINKIN, which is found throughout Metazoa and have demonstrated that LINKIN functions as an adhesion molecule. LINKIN is an apically and laterally enriched transmembrane protein that is expressed on the surface of interconnected cells in the *C. elegans* male gonad. Without this protein there are cell–cell adhesion defects during the collective migration of the male gonad. In the extracellular domain, we identified seven atypical FG–GAP domains that likely fold into a β-propeller structure, which is similar to the ligand-binding domain of α-integrins. We have also identified three intracellular interactors of LINKIN—RUVBL1, RUVBL2, and

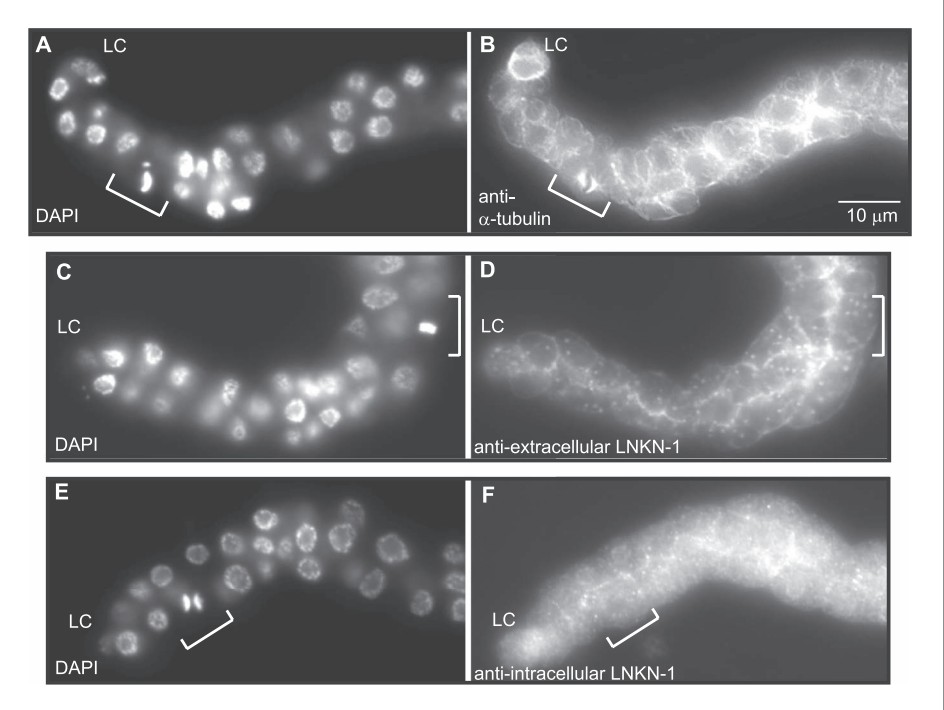

**Figure 8**. LNKN-1 remains at the plasma membrane during cell division. (**A**, **C**, **E**) In DAPI-stained dissected male gonads, dividing cells (white bracket) were identified by their condensed chromosomes. (**B**) Anti-α-tubulin staining shows that microtubules redistribute during cell division to radiate out from the mitotic spindle. (**D** and **F**) Staining with anti-LNKN-1 antibody against either the extracellular domain (**D**) or intracellular domain (**F**) shows that LNKN-1 remains at the membrane during cell division. LNKN-1 localization is the same in dividing cells (white bracket) and non-dividing neighbors.

α-tubulin. β-tubulin may also be an interactor based on mass spectrometry identification, *C. elegans* gonadal adhesion function, and known interaction to α-tubulin, but we were unable to confirm β-tubulin interaction by immunoprecipitation/Western blot analysis. We propose that LINKIN interacts with the microtubule cytoskeleton, and this interaction is modulated by RUVBL1 and RUVBL2 AAA+ ATPases.

## LINKIN protein family is conserved in Metazoa

We have demonstrated that LINKIN is a unique transmembrane glycoprotein that functions as an adhesion molecule. This approximately 600 AA protein has a large extracellular domain of approximately 530 AA and a short 22 AA intracellular domain. We found a number of notable features about LINKIN. First, it is a protein pre-dating metazoans. Second, the size of the protein and organization of the domains have not changed during its evolution. The seven atypical FG–GAP domains, the extracellular region proximal to the transmembrane domain, and the intracellular sequence are all conserved. The identical sequence at the C-terminal of LINKIN suggested that some of its intracellular interactors may also be conserved proteins. In fact, LINKIN interactors, RUVBL1, RUVBL2, and α-tubulin are all highly conserved proteins that likely pre-date LINKIN. Third, among the genomes we searched, including those of *H. sapiens*, *M. musculus*, *D. melanogaster*, *C. elegans*, and *T. adhaerens*, LINKIN has no paralog. Even the highly conserved short 22 AA intracellular domain is not found in other proteins. The lack of paralogs is surprising considering that other highly conserved adhesion molecules like integrins and cadherins have numerous paralogs and have expanded into superfamilies (*Angst et al., 2001*; *Takada et al., 2007*).

The presence of seven atypical FG–GAP domains in the extracellular domain suggests that LINKIN might fold into a seven-bladed β-propeller. The significance of this prediction is that this structure resembles the ligand-binding domain of the adhesion molecule α-integrin. As with α-integrins, the β-propeller structure for LINKIN is located towards the amino-terminal of the extracellular domain.

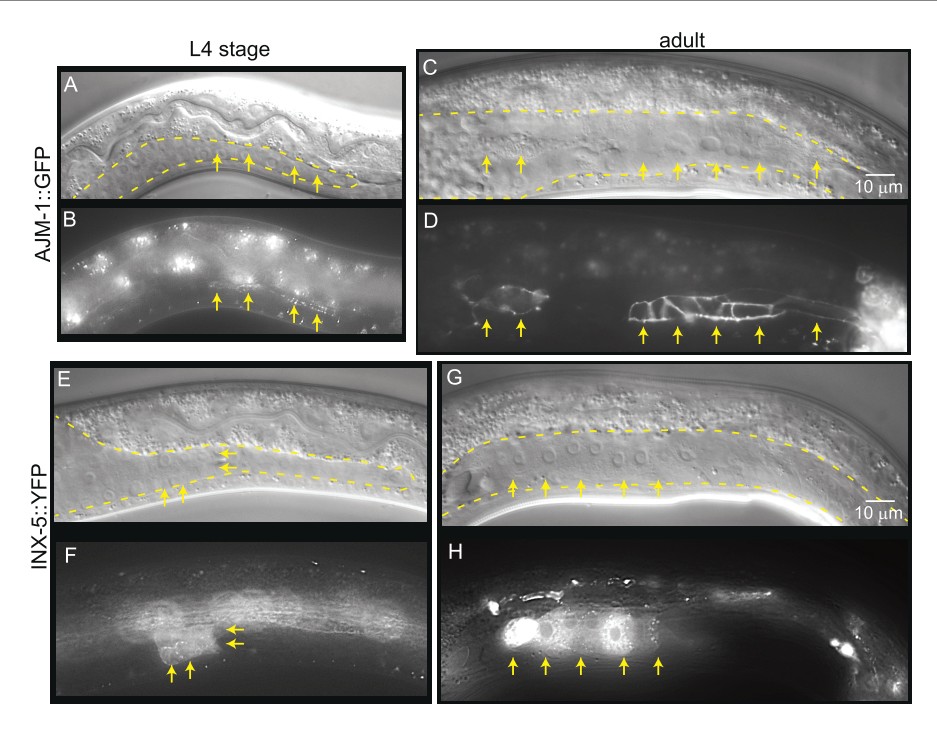

**Figure 9**. Mature cell–cell junctions form during the L4 stage. (**A** and **B**) Adherens junction marker, AJM-1::GFP, begins to localize as puncta to the apical region of the gonad in the L4 stage (arrows). (**C** and **D**) In the adult gonad, AJM-1::GFP lines the apical junctions. (**E** and **F**) INX-5::YFP, an innexin expressed in the male gonad, begins to be expressed and accumulate at gap junctions in the L4 stage in a cluster of somatic gonadal cells (arrow). (**G** and **H**) INX-5::YFP expression becomes stronger in the adult. Neither AJM-1::GFP nor INX-5::YFP expresses in the gonad in the L3 stage. For each image pair, the top panel is a Nomarski image and bottom is a fluorescence image.

LINKIN is expressed on the surface of each adherent cell and might use the β-propeller domain to bind ligands on a neighboring cell to promote adhesion.

## LNKN-1 localizes to apical and lateral plasma membrane and functions in cell adhesion

The *C. elegans* gonad shape develops through a collective migration of epithelial-like cells lead by the LC. One of the striking phenotypes of the *lnkn-1* mutant was cell detachment in the migrating gonad. In *lnkn-1* mutants, gonadal cell dissociation occurs in the L3 stage. At this stage, although the adhesions are strong enough that these cells cannot be mechanically dissociated without lysing, they have not yet formed mature adherens junctions. The expression of adhesion junction markers, AJM (apical junction molecule)-1::GFP, and gap junction marker, INX(innexin)-5::GFP, was absent in the L3 stage, when gonad cells begin detaching in *lnkn-1* mutants; their expression only begins in the L4 stage and grows stronger in the fully differentiated adult gonad. The role of adhesion molecules, such as LINKIN, may therefore be more important before other, possibly stronger, adhesions are formed. The gonad detachment defect of *lnkn-1* mutants likely results from the combined effects of the absence of a functional LINKIN adhesion molecule, the lack of other permanent adhesion structures in the L3 stage, and the force generated by the migrating LC.

LNKN-1 was present in many *C. elegans* tissues that possess apical/basal polarity, including the intestine, excretory canal, vulva, hook cells, and gonad; its localization was stronger on the apical and lateral sides for the tissues examined. YFP-tagged to either the extracellular or intracellular domain of LNKN-1 changes its localization from apical- and lateral-biased to uniform plasma membrane localization, indicating that apical localization is an active process requiring the function of both extracellular and intracellular domains of the protein. YFP-tagging also disrupts the function of LINKIN, as neither YFP-tagged LNKN-1 construct rescued the mutant phenotype, suggesting that apical localization may

be necessary for LNKN-1 function. Known adhesion molecules have preferential localization at the plasma membrane. Integrins are often enriched in the basal domain since they bind extracellular matrix (*Schoenenberger et al., 1994*), while cadherins are enriched in lateral domains (*Halbleib and Nelson, 2006*). Specialized adhesion molecules like claudins and occludins localize to tight junctions (*González-Mariscal et al., 2003*) and connexins to gap junctions (*Evans and Martin, 2002*). LINKIN may be an adhesion molecule for cell–cell contacts and apical junctions.

Our rescue experiments also showed that secreted forms and partial domains of LNKN-1 do not provide function. The only previous study of LINKIN showed that the extracellular domain of human LINKIN functions as a secreted protein to modulate T-cell activation in cell culture and graft-versus-host disease model (*Fiscella et al., 2003*). Our experiments indicate that in the context of *C. elegans* collective migration, a secreted extracellular domain is insufficient to rescue the detachment defect. Our results suggest that, in the many *C. elegans* and human tissues expressing LINKIN, its function could originally have been as a transmembrane adhesion molecule, which in vertebrates conceivably has expanded to include a secreted form.

## Functions for LINKIN interactors, RUVBL proteins, and microtubules

Although we do not yet know the binding partner on the extracellular side, we have made progress in identifying conserved interactors on the intracellular side—RUVBL1, RUVBL2 and α-tubulin. RUVBL1 and RUVBL2 are highly conserved members of the AAA+ ATPase superfamily, which use the energy harvested from ATP hydrolysis to perform mechanical functions on macromolecules. In bacteria where they were first identified, RUVBs have a function as a DNA helicase at holiday junctions; but in more complex organisms, the RUVBLs have additional diverse functions (*Jha and Dutta, 2009*). Among its non-nuclear roles, RUVBLs function in R2TP co-chaperone complex assembly (*Kakihara and Houry, 2012*) and in spindle assembly by nucleating microtubules and localizing components to the mitotic spindle (*Gartner et al., 2003*; *Ducat et al., 2008*). We have shown through limited cell fractionation that LINKIN is present in the membrane, where it interacts with RUVBL proteins and α-tubulin. Although we showed that LINKIN does not localize to the mitotic spindle, the latter role of RUVBLs as regulators of microtubule assembly and complex formation may be most relevant to its function with LINKIN and α-tubulin. Based on previously known RUVBL functions, we propose that RUVBL1 and RUVBL2 form a heterometric ring structure that promotes assembly of a LINKIN complex and nucleation of microtubules (*Figure 10*).

Most adhesion receptors interact with the cell cytoskeleton through their intracellular domain (*Juliano, 2002*); LINKIN interacts with microtubules. Microtubules are known to play important functions in cell migration and tissue organization (*Gauthier-Rouvière et al., 2004*; *Etienne-Manneville, 2013*). They provide structure and rigidity to tissues so that they can withstand high compression forces (*Brangwynne et al., 2007*), and the bundled cortical microtubules of gonadal cells likely provide such a structure. Microtubules are also involved in creating polarity and trafficking components along polarized tracks. The subunits have a plus and minus end polarity growing out from the centrosome, which in turn can create a front–back polarization in migratory cells through selective stabilization of fibers (*Wadsworth, 1999*) and polarized trafficking to the membrane (*Miller et al., 2009*). Microtubules serve as tracks to transport cadherin-containing vesicles to specific areas of the plasma membrane to establish cell adhesion (*Mary et al., 2002*). LINKIN may serve to anchor the microtubule cytoskeleton to particular domains of the plasma membrane. Conversely, microtubules and microtubule-associated motor proteins may transport LINKIN to select domains of the plasma membrane, helping to establish LINKIN localization to apical domains and cell–cell contacts.

As investigations into other cell adhesion molecules have revealed numerous important functions in animal development (*Thiery, 2003*; *Halbleib and Nelson, 2006*), LINKIN's expression in many human and *C. elegans* tissues suggests that future studies will demonstrate its involvement in many processes. We have demonstrated an adhesion function for the transmembrane glycoprotein LINKIN and have identified interactors RUVBL1, RUVBL2, and α-tubulin, which support a model for LINKIN regulating the microtubule cytoskeleton. Considering that the interactions between LINKIN, RUVBL1, RUVBL2, and α-tubulin were identified using a human cell line and were also required for gonad cell adhesion in a nematode worm, these interactions may have a conserved molecular function in Metazoa including human. We are proposing that LINKIN functions in maintaining tissue integrity through cell–cell adhesion and apically polarization, but further studies are necessary to show the generality of this function and to elucidate differences between the roles of LINKIN and other adhesion receptors.

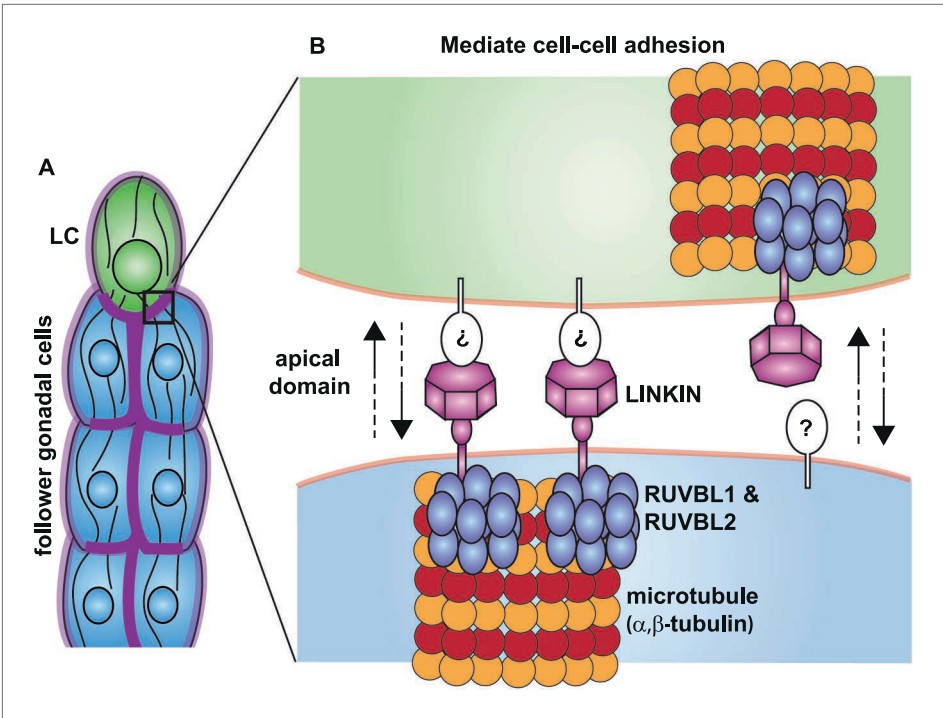

**Figure 10**. Model for the function of LINKIN, RUVBL1, RUVBL2, and microtubule proteins in cell–cell adhesion. (**A**) The male gonad shape is generated by a collective migration of the leader LC (green) and follower somatic cells (blue). LINKIN (purple) is a transmembrane glycoprotein expressed in the plasma membrane of all gonadal cells, with enrichment at apical and lateral domains (dark purple). LNKN-1 is required for cell–cell adhesion during gonadal migration. (**B**) The interface between two adherent cells boxed in (**A**) is shown in more detail. LINKIN integrates interactions with neighboring cells on the cell exterior with connections to the microtubule cytoskeleton on the cell interior. On the extracellular side, the β-propeller domain (purple heptagon) of LINKIN binds an unidentified partner (white oval) on the adjacent cell membrane. The highly conserved intracellular domain of LINKIN binds RUVBL1, RUVBL2, and α-tubulin at the intracellular face of the plasma membrane. Based on RUVBL1 and RUVBL2's ability to form stacked hetero-hexameric rings (**Gorynia et al., 2011**) and regulate microtubule nucleation and dynamics (**Gartner et al., 2003**), we propose that they assist in interaction between LINKIN and microtubules.

## Materials and methods

### Worm strains

*C. elegans* strains were cultured at room temperature using standard protocols unless indicated otherwise (**Brenner, 1974**). Strain VC877 *tag-256(gk367)/hT2* was obtained from the Caenorhabditis Genetics Center (CGC). Other alleles and transgenes used in this study are *him-5(e1490)* (**Hodgkin et al., 1979**) | PS4730 *syIs128 [lag-2::YFP]; him-5(e1490)* (**Kato and Sternberg, 2009**) | *syIs78 [ajm-1::GFP]* (**Gupta et al., 2003**) | *inx-5::GFP* | PS6018 *unc-119(ed4); him-5(e1490) syEx1130[tag-256::TAG-256::YFP(20 ng/μl) + unc-119(+) (70 ng/μl) + Bluescript]* | PS6372 *tag-256(gk367); him-5(e1490); syEx1184[tag-256::TAG:256(5 ng/μl) + myo-2::mCherry]*.

### *lnkn-1* cDNA production from *lnkn-1(gk367)* mutant

Total RNA was extracted from 25 *lnkn-1(gk367)* mutant animals using a TRIzol extraction method (**Chomczynski and Sacchi, 1987**) but modified for small quantities of worms (Morimoto group, http://groups.molbiosci.northwestern.edu/morimoto/research/Protocols/IX.%20C.%20elegans/B.%20Extraction/2.%20TotalRNA.pdf). This was followed by a reverse transcription reaction using SuperScriptIII first-strand synthesis supermix (Invitrogen, Waltham, MA) following manufacturer's instructions and PCR using *lnkn-1* primers to the beginning and end of the gene. The amplified product was submitted for DNA sequencing (Laragen).

## Plasmid constructions for *lnkn-1* rescue constructs

See *Supplementary file 3*.

## Generating transgenic animals

For *lnkn-1* mutant rescue experiments, a DNA mixture of 5 ng/µl or 1 ng/µl of *lnkn-1* rescuing construct, 7 ng/µl of *myo-2::dsRed*, and 150 ng/µl of 1 kb ladder (NEB) as carrier DNA, was injected into adult gonads of PS6372 *lnkn-1(gk367)/hT2*; *him-5(e1490)* hermaphrodites.

## RNAi feeding assays

Genecards.org was used to identify the *C. elegans* homologs of the human genes identified by mass spectrometry. *C. elegans* genes were screened using an RNAi protocol previously described (*Kamath et al., 2001*), with a few modifications. Single RNAi bacterial colonies were grown for 6–8 hr in LB with carbenicillin selection (100 µg/ml). Carbenicillin (25 µg/ml) and IPTG (1 mM) were spread on NGM plates just prior to adding 200 µl of RNAi bacterial culture, and plates were dried at RT overnight. The following day, eggs were harvested from gravid adults by bleaching and placed on plates containing RNAi bacteria. Animals were grown at RT and L4 stage males were scored by Nomarski and fluorescence microscopy for detached gonad. The RNAi bacteria were obtained from the Vidal library (*Rual et al., 2004*) when the gene was available, and the Ahringer library (Kamath et al., 2003) otherwise.

## Mass spectrometry analyses

### Immunoprecipitation

Two HEK 293T cultures were grown, one culture labeled with light amino acid and the other labeled with heavy arginine and lysine (Arg6 and Lys8). ITFG1-Myc protein was transiently expressed after 24 hr of incubating with Fugene HD transfection reagent and ITFG1-Myc expression plasmid (RC204773, Origene, Rockville, MD) in light amino acid culture. Control heavy cells were incubated with Fugene HD transfection mixture without any plasmid. Cells were washed twice with 15 ml cold PBS and lysed in lysis buffer (50 mM HEPES, pH 7.5; 70 mM KOAc; 5 mM Mg(OAc)2; 20 0.2% n-dodecyl-β-D-maltoside, 10 µM bortezomib, 1 mM N-ethylmaleimide, one tablet of complete protease inhibitor [Roche, Germany]) for 30 min on a rotator at 4°C. Cell lysate was cleared by centrifugation for 20 min at 13,000 rpm in an Eppendorf centrifuge (5417R), and supernatant was transferred to a new tube. Protein concentrations for both cell lysate were determined by using Bradford reagent. For each IP, Myc-beads (80 µl, Sigma) were washed twice with 1 ml lysis buffer. Clear cell lysate (8 mg) was incubated with Myc-beads at 4°C for 1 hr. Beads were washed with 1 ml lysis buffer three times and 100 mM Tris buffer (pH 8.5) twice. Proteins were eluted with 45 µl of 10 M urea in 100 mM Tris buffer (pH 8.5) by incubating at 37°C for 15 min by using Bio-rad micro bio-spin chromatography column. 32 µl of light IP was mixed with 32 µl of heavy IP for Mass Spectrometric Analysis. The remaining 13 µl was diluted with 87 µl of water and 33 µl of 4× SDS sample buffer for Western blot analysis.

### In-solution tryptic digest followed by Nano-LC-MS/MS analysis

Eluted light and heavy proteins (64 µl) were diluted with 16 µl of 100 mM Tris buffer (pH 8.5) and reduced with 0.5 µl of 0.5 M Tris(2-carboxyethyl)phosphine hydrochloride (Thermo Fisher Scientific, Waltham, MA) for 20 min at 37°C, alkylated with 1.8 µl of 0.5 M chloroacetamide (Thermo Fisher Scientific) for 15 min at 37°C, and digested with 2 µl of 100 ng/µl lysyl endopeptidase (Lys-C, Wako Chemicals, Richmond, VA) for 4 hr at 37°C. Samples were diluted to a final concentration of 2 M urea by adding 240 µl of 100 mM Tris–HCl pH 8.5 and digested with 32 µl of 100 ng/µl trypsin (Thermo Fisher Scientific) and 3.2 µl 100 mM CaCl$_2$ for 18 hr at 37°C. After desalting with a Vivapure C18 micro spin column (Sartorius Stedim Biotech, Bohemia, NY), peptides were eluted with 50 µl of 75% acetonitrile and 0.2% formic acid twice. Solvent was removed using a SpeedVac. Dried samples were acidified by 0.2% formic acid and loaded onto an Easy Nano-LC connected to a hybrid LTQ-Orbitrap Classic (Thermo Fisher Scientific) as previously described (*de Godoy et al., 2008*; *Lee et al., 2011*).

### Mass spectrometry data analysis

Peak lists were generated from raw data files using MaxQuant (version 1.0.13.13) as described previously (*Lee et al., 2011*). Data were analyzed by combining two raw data files from two mass spectrometer analyses. We only considered Light/Heavy ratio that can be quantified from more than two peptide counts with one unique peptide.

## Western blot analysis

Equal amounts of cell lysate were mixed with SDS sample buffer (4×) and heated at 90°C for 5 min. After centrifugation at 3000×*g* for 30 s, sample (25 µg) was loaded on a 4–12% or 4–20% SDS-PAGE. Proteins were transferred to nitrocellulose membranes and stained with Ponceau S. The nitrocellulose membranes were blocked with 5% milk/TBST for 5 min to remove Ponceau S and for additional 30 min. Primary antibodies were prepared in 3% milk/TBST and incubated for 1 hr at room temperature on a shaker. After removing the primary antibody, membranes were washed three times with 3% milk/TBST for 5 min each. Primary antibodies were used including rabbit polyclonal antibodies against human RUVBL1 and RUVBL2 (Proteintech), anti-β-tubulin and anti-β-actin. Secondary antibodies were incubated for 1 hr at room temperature on a shaker. Membranes were washed three times with TBST with 5 min each time. ECL Plus (GE healthcare) was used to detect signal.

## Cell fractionation

HEK 293T cells were transfected with plasmids encoding ITFG1-Myc (RC204773, Origene), Flag-RUVBL1 (51635, Addgene, Cambridge, MA), and HA-RUVBL2 (51636, Addgene) or with control Myc-vector. Mem-PER Plus Membrane Protein Extraction Kit (Thermo Fisher Scientific) was used to isolate membrane and cytoplasmic fractions from harvested cells. Flag-RUVBL1 was immunoprecipitated from the two fractions using Flag-beads (Sigma-Aldrich, St. Louis, MO). Immunoprecipitation/Western blot analysis was performed as described above.

## *C. elegans* antibody production

A soluble LNKN-1 protein containing the entire extracellular domain was expressed and purified from S2 cells by the Caltech protein expression facility. Rabbit polyclonal antibodies were generated against this LNKN-1 extracellular domain protein and a 17 AA intracellular domain peptide, Ac-C(Ahx) DRYERQQQSHRFHFDAM-OH (QCB, Hopkinton, MA). The antibodies were affinity-purified using their antigens, either the extracellular domain protein or the intracellular domain peptide. Rabbit polyclonal antibodies were also generated against the entire RUVB-1 and RUVB-2 proteins (Proteintech) and affinity-purified using the RUVB-1 and RUVB-2 proteins. Specificity of all antibodies was tested by staining tissues from *C. elegans* that were either treated with RNAi against the antigen gene or mutant for the gene. Reduction in staining was observed for each of these antibodies (*Figure 3— figure supplement 1*, *Figure 6—figure supplement 1*).

## Antibody staining of *C. elegans* gonads

Gonads were dissected from larval stage males following Chan and Meyer's 'Protocol 21: Antibody staining of *C. elegans* gonads' (*Shaham, 2006*). A final concentration of 2% paraformaldehyde was used and phosphate buffered saline was substituted for sperm salts. Primary antibodies were used at 1:250 dilution for antibody against LNKN-1 extracellular domain, RUVB-1, and RUVB-2 and at 1:200 dilution for LNKN-1 intracellular domain. Mouse monoclonal antibodies against α-tubulin (12G10, supernatant, Developmental Studies Hybridoma Bank, Iowa City, IA) and DLG-1 (DLG-1, supernatant, DSHB) were used at a 1:100 dilution. Secondary antibodies against rabbit (Alexa Fluor 594 Goat Anti-Rabbit IgG, Life Technologies, Carlsbad, CA) and mouse (Alexa Fluor 594 Goat Anti-Mouse IgG, Life Technologies) were used at 1:500 dilution. Tissues were mounted on slides using Vectashield mounting media containing DAPI (Vector Laboratories, Burlingame, CA).

## Sequence alignment and domain assignments

LINKIN sequences from *H. sapiens*, *M. musculus*, *D. melanogaster* (uniprot.org), *C. elegans* (wormbase.org) were aligned using clustalw (http://www.genome.jp/tools/clustalw/) and clustalo (http://www.ebi.ac.uk/Tools/msa/clustalo/). *T. adhaerens* LINKIN is TRIADDRAFT_52570 but the gene prediction is imperfect (metazoa.ensembl.org).

## Identifying FG–GAP calcium-binding domain motif

The FG–GAP domain, first identified in α-integrin, contains a loosely conserved sequence of Phe-Gly and Gly-Ala-Pro. A motif frequently found between the FG and GAP sequences is a DxDxDG calcium-binding motif (D = Asp, G = Gly, x = AA; *Chouhan et al., 2011*). While DxDxDG is a common calcium-binding motif, variations on this motif depend on the particular protein family (*Rigden and Galperin, 2004*). A comparison of all human α-integrin FG–GAP domains showed that their calcium-binding

motif has a strong D̲xxxD̲xxxD signature, which contains a more weakly conserved D̲xD̲xD̲G sequence as its first 6 AAs (*Chouhan et al., 2011*). This D̲xxxD̲xxxD signature is different from DxDxDG calcium-binding regions of other proteins like the EF-hand protein family (*Rigden and Galperin, 2004*). The FG–GAP sequence was only loosely conserved and not always identifiable, but the DxxxDxxxD calcium-binding domain was highly conserved in LINKIN.

## Acknowledgements

We are grateful to Amir Sapir, Hillel Schwartz, and Srimoyee Ghosh for critical reading of the manuscript and to TFC's postdocs, Xiaoyi Zhaang and Lin Gui, for assistance with IP/Western blot analysis experiments. Caltech protein expression facility expressed and purified LNKN-1 extracellular domain. Mass spectrometry experiments were performed at the Caltech protein exploration laboratory directed by Dr Sonja Hess and Prof Ray Deshaies. Wormbase (wormbase.org) was a valuable resource for *C. elegans* information as was the British Columbia *C. elegans* Gene Expression Consortium (http://elegans.bcgsc.ca/home/ge_consortium.html) GFP expression pattern database. The monoclonal antibody developed by Frankel J and Nelsen EM was obtained from the Developmental Studies Hybridoma Bank, created by the NICHD of the NIH, and maintained at The University of Iowa. Nematode strains were provided by the *Caenorhabditis* Genetics Center, which is funded by the NIH National Center for Research Resources. CZY was funded by a CIRM pre-doctoral training grant. PWS is an investigator with the Howard Hughes Medical Institute, which supported this work.

## Additional information

### Funding

| Funder | Grant reference number | Author |
| --- | --- | --- |
| Howard Hughes Medical Institute | 047-101 Paul W Sternberg | Paul W Sternberg |
| California Institute for Regenerative Medicine | Collin Z Yu | Collin Z Yu |

The funders had no role in study design, data collection and interpretation, or the decision to submit the work for publication.

### Author contributions

MK, T-FC, Conception and design, Acquisition of data, Analysis and interpretation of data, Drafting or revising the article; CZY, JD, Acquisition of data, Analysis and interpretation of data; PWS, Conception and design, Analysis and interpretation of data, Drafting or revising the article

### Author ORCIDs

Paul W Sternberg, http://orcid.org/0000-0002-7699-0173

## Additional files

### Supplementary files

• Supplementary file 1. Interactors of human LINKIN identified by mass spectrometry.

• Supplementary file 2. Top interactors of human LINKIN identified by mass spectrometry (>fivefold enriched over control).

• Supplementary file 3. *lnkn-1* constructs used for rescuing *lnkn-1* mutant.

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
