## [Decision Letter]

Thank you for sending your work entitled “LINKIN, a new transmembrane protein necessary for cell adhesion” for consideration at *eLife*. Your article has been evaluated by Janet Rossant (Senior editor) and 2 reviewers.

The reviewers and the editor had an extensive discussion on this paper. All of us were intrigued by the identification of LINKIN as a member of new class of potentially conserved adhesion proteins. It was also agreed the work demonstrating a role for LINKIN in migration in *C. elegans* was convincing. However major flaws were identified in your molecular evidence for the LINKIN-RuvB complex and in your interpretation of the immunolocalization of the different components. If you wish to claim that this complex is important, more direct in vivo evidence (beyond IP/MS) is required for the complex proposed: both reviews point out weaknesses in this part of the paper.

In addition, we felt that you had drastically over-interpreted the functional significance of the phylogenetic distribution of LINKIN proteins. This part of the paper would need to be rewritten in a more cautious manner.

Although these are serious concerns, the novelty of the findings is attractive and we would encourage you to try to address the concerns raised in a careful revision of the paper and we will be happy to reconsider a revised version. The reviewers' comments are below for your responses.

*Reviewer* #*1*

The authors argue that LINKIN physically interacts with RUBVL1, RUVBL2, and alpha tubulin, based on mass spectrometry data and co-immunopreciation from a myc tagged LINKIN protein. To support this conclusion, the authors should either provide binding affinities for these interactions or perform reciprocal pulldowns with the hypothesized interaction partners. The localization of RUVBL1 and RUVBL2 is also not consistent with a direct interaction.

In the Discussion section and elsewhere, the authors draw unsupported inferences about the function of LINKIN based on its phylogenetic distribution. For example, the statement, “First, it is a protein pre-dating metazoans, suggesting that its function is fundamental to multicellular organization and may even have contributed to the evolution of multicellular life, as maybe the case with the integrin and cadherin family of adhesion molecules (Abedin and King, 2008; Sebé-Pedrós et al., 2010).” Many proteins pre-date metazoan origins, including actin, tubulin, DNA polymerase, etc. The fact that these proteins evolved before animals does not indicate that they are fundamental to multicellular organization. For that, it is necessary to show that LINKIN has a conserved role in adhesion in diverse animals and their closest relatives. In the next sentence, the authors state that “LINKIN is present in Trichoplax, an animal of the most basal phylum Placozoa that is composed of two epithelioid cell layers, fitting with a cell-cell adhesion function for LINKIN.” Again, the presence of a gene in a multicellular organism is not sufficient to infer that the gene plays a role in cell adhesion. There are a multitude of other cellular functions that could be served. The authors should exercise more caution in making inferences about LINKIN function based on its phylogenetic distribution.

*Reviewer* #*2*

Mass spec: These data are inadequately described. It would in any case have made more sense to label the LINKIN-Myc cultures with heavy isotope, such that any protein elevated in that sample relative to controls would be definitively cell-derived. The analysis is not given in sufficient detail; Supplementary Table 2 simply indicates >5-fold enriched (based on 2 peptides). That's okay but what are the two columns giving relative enrichments of the proteins listed in Supplementary Table 3: one labeled counts and the other is rank-ordered (presumably by isotope ratio). Then Figure 5 lists “relative counts”, I think those should be the numbers from the first rank-ordered column of Supplementary Table 3 and it looks from the figure as if they are (except that LINKIN is listed here as 33-fold enriched although Supplementary Table 3 lists it as 193-fold enriched, if I read it correctly.

So what is the “counts” column in Supplementary Table 3: is it spectra, or peptides perhaps?

Several other columns in Supplementary Table 3 are not explained or labeled.

This whole analysis needs a better explanation to allow one to evaluate it. The many different tubulins listed in the table, several of which out-rank at least one of the RuvB proteins are not discussed; why concentrate only on tub a?

The evidence for the association of LINKIN/RuvBs and tubuin could be strengthened. Do RUVBLs IP with α-tubulin in cells with or without ITFG knockdown? Does LINKIN knockdown impact localization of RUVBLs and a-tubulin? The authors should also include (or at least discuss) some other prominent cytoskeleton protein that fails to bind to ITFG as a negative control in their co-IP experiments.

Some obvious further experiments on the proposed RuvB/tubulin complex with LINKIN are not included. For example, the authors show that LINKIN precedes some other junctional proteins but what about the proposed members of the complex? What is the localization of different components of the LNKN-1 complex at L3 stage when cell junctions are being rearranged? Do they colocalize at the cellular membrane?

The interpretation of the IF data on the putative interactors is overstated. The widespread cytoplasmic distribution of the RuvB proteins (Figure 6) certainly cannot be interpreted as “suggesting that RuvBs could function in the cytoplasmic face of the plasma membrane”; they may, but these data do not “suggest” that. Similarly the retention of LINKIN at the membrane during mitosis does not “further support LNKN-1 interacting with RUVBLs and microtubules at the plasma membrane”; the manuscript provides no such support.

Also, it is claimed in the manuscript text that it demonstrates that the interactions are “functionally conserved in nematodes”, bit of an overstatement there too.

---

## [Author Response]

Reviewer #1

*The authors argue that LINKIN physically interacts with RUBVL1, RUVBL2, and alpha tubulin, based on mass spectrometry data and co-immunopreciation from a myc tagged LINKIN protein. To support this conclusion, the authors should either provide binding affinities for these interactions or perform reciprocal pulldowns with the hypothesized interaction partners. The localization of RUVBL1 and RUVBL2 is also not consistent with a direct interaction*.

We have performed a reciprocal immunoprecipitation with Flag-RUVBL1 and have used Western blot analysis to show that LINKIN, RUVBL2, and α-tubulin interact. We also fractionated the membrane and cytoplasmic compartments from cell lysates and found that the interactions of LINKIN, RUVBL1, RUVBL2, and α-tubulin occur in the membrane fraction. RUVBL1, RUVBL2, andα-tubulin also interacted in the cytoplasm without LINKIN. These results have been added to Figure 5 and are described in the manuscript text. This data together with the cytoplasmic localization of RUVB proteins in *C. elegans* male gonad supports our hypothesis that RUVB interacts with the cytoplasmic domain of LINKIN at the plasma membrane. We attempted to perform reciprocal IP/Western blots at endogenous levels with RUVBL1 and RUVBL2 also but were unsuccessful with the immunoprecipitation, probably due to the antibodies affecting the interactions or to the quality of the antibodies for IP.

*In the Discussion section and elsewhere, the authors draw unsupported inferences about the function of LINKIN based on its phylogenetic distribution. For example, the statement, “First, it is a protein pre-dating metazoans, suggesting that its function is fundamental to multicellular organization and may even have contributed to the evolution of multicellular life, as maybe the case with the integrin and cadherin family of adhesion molecules (Abedin and King, 2008; Sebé-Pedrós et al., 2010).” Many proteins pre-date metazoan origins, including actin, tubulin, DNA polymerase, etc. The fact that these proteins evolved before animals does not indicate that they are fundamental to multicellular organization. For that, it is necessary to show that LINKIN has a conserved role in adhesion in diverse animals and their closest relatives. In the next sentence, the authors state that “LINKIN is present in Trichoplax, an animal of the most basal phylum Placozoa that is composed of two epithelioid cell layers, fitting with a cell-cell adhesion function for LINKIN.” Again, the presence of a gene in a multicellular organism is not sufficient to infer that the gene plays a role in cell adhesion. There are a multitude of other cellular functions that could be served. The authors should exercise more caution in making inferences about LINKIN function based on its phylogenetic distribution*.

We have removed or rewritten sentences that draw untested inferences based on phylogeny.

Reviewer #2

*Mass spec: These data are inadequately described. It would in any case have made more sense to label the LINKIN-Myc cultures with heavy isotope, such that any protein elevated in that sample relative to controls would be definitively cell-derived. The analysis is not given in sufficient detail; Supplementary Table 2 simply indicates >5-fold enriched (based on 2 peptides). That's okay but what are the two columns giving relative enrichments of the proteins listed in Supplementary Table 3: one labeled counts and the other is rank-ordered (presumably by isotope ratio). Then*
Figure 5
*lists “relative counts”, I think those should be the numbers from the first rank-ordered column of Supplementary Table 3 and it looks from the figure as if they are (except that LINKIN is listed here as 33-fold enriched although Supplementary Table 3 lists it as 193-fold enriched, if I read it correctly*.

*So what is the “counts” column in Supplementary Table 3*: *is it spectra, or peptides perhaps?*

*Several other columns in Supplementary Table 3 are not explained or labeled*.

*This whole analysis needs a better explanation to allow one to evaluate it*.

For Supplementary Table 1, we have removed extraneous columns, renamed column headings, and added heading descriptors (column N and O) to facilitate reading the table. The candidate interactor proteins that we tested were those that were more than 5-fold abundant in the LINKIN experiment over control experiment (column C). LINKIN (ITFG1) is 35-fold enriched (column C, row 9), while the 193-fold increase corresponds to the first gene on the list. “Counts” is peptide types and we have renamed this column.

*The many different tubulins listed in the table, several of which out-rank at least one of the RuvB proteins are not discussed; why concentrate only on tub a*?

We chose our *C. elegans* genes based on homology to the human genes identified by mass spectrometry. The homologous *C. elegans* α-tubulin to the one identified by mass spectrometry yielded a phenotype. Because the homologous α-tubulin did not yield a phenotype, we tested other *C. elegans* α-tubulins and found that *tbb-2* (RNAi) produces a gonad detachment phenotype. This result has been added to the text and to Figure 5. Not all tubulins produced the gonadal effect, perhaps because not all are expressed in the linker cell. By Western blot analysis, we did not detect α-tubulin from LINKIN-myc immunoprecipitates. This has been added to the text and to Figure 5.

*The evidence for the association of LINKIN/RuvBs and tubuin could be strengthened. Do RUVBLs IP with α-tubulin in cells with or without ITFG knockdown? Does LINKIN knockdown impact localization of RUVBLs and a-tubulin? The authors should also include (or at least discuss) some other prominent cytoskeleton protein that fails to bind to ITFG as a negative control in their co-IP experiments*.

We have performed additional IP/Western blot analyses to better characterize the interaction between LINKIN, RUVBL1, RUVBL2, and α-tubulin. We had already shown that by immunoprecipitating LINKIN, we see interactions to the other three proteins by Western blot analysis.

We have now also immunoprecipitated using RUVBL1 and performed Western blot analysis to show that RUVBL1 interacts with the other three proteins. This result is described in the text under the section title, “LINKIN binds RUVBL1, RUVBL2, and α-tubulin at the plasma membrane”. For this experiment, we separated the cytoplasmic and the membrane fractions, and showed that the interaction of RUVBL1 with LINKIN, RUVBL2, and α-tubulin occurred only in the membrane (Figure 5), while the interaction between RUVBL1, RUVBL2 and α-tubulin also occurred in the cytoplasm without LINKIN (Figure 5). The ability of RUVBL1, RUVBL2, and α-tubulin to interact without LINKIN is consistent with other reports of this interaction occurring in the cytoplasm during cell division (19; 12; 11).

We have performed a cytoskeleton protein control using β-actin. By immunoprecipitating with Myc-LINKIN under identical conditions as previously and performing a Western blot analysis with anti-β-actin antibody, we show that this abundant cytoskeletal protein does not immunoprecipitate with LINKIN. This control has been added to Figure 5.

*Some obvious further experiments on the proposed RuvB/tubulin complex with LINKIN are not included. For example, the authors show that LINKIN precedes some other junctional proteins but what about the proposed members of the complex? What is the localization of different components of the LNKN-1 complex at L3 stage when cell junctions are being rearranged? Do they colocalize at the cellular membrane*?

Because the RUVBL proteins and α-tubulin have a strong cytoplasmic localization throughout gonadal development, and the difference between the cytoplasmic and membrane populations cannot be detected by staining, we could not assess when they first accumulate at the membrane. We did however investigate whether RNAi knockdown of RUVB-1, RUVB-2, or α-tubulin affects either the expression or plasma membrane localization of LINKIN, since RUVB proteins have been reported to act as transcriptional regulators, and microtubules in protein transport. None of these LINKIN interactors affected LINKIN expression or localization. This result has been added to Figure 7 and to the text.

*The interpretation of the IF data on the putative interactors is overstated. The widespread cytoplasmic distribution of the RuvB proteins (*Figure 6*) certainly cannot be interpreted as “suggesting that RuvBs could function in the cytoplasmic face of the plasma membrane”; they may, but these data do not “suggest” that. Similarly the retention of LINKIN at the membrane during mitosis does not “further support LNKN-1 interacting with RUVBLs and microtubules at the plasma membrane”; the manuscript provides no such support*.

We have removed these claims. To address the issue of whether the interaction occurs at the plasma membrane, we fractionated the membrane and cytoplasmic layers from HEK293T cells before performing an IP/Western blot analysis using Flag-RUVBL1 to immunoprecipitate (described above). We showed that LINKIN, RUVBL1, RUVBL2, and α-tubulin interact at the membrane (Figure 5).

*Also, it is claimed in the manuscript text that it demonstrates that the interactions are “functionally conserved in nematodes”, bit of an overstatement there too*.

We have rewritten the sentence to: “Considering that the interactions between LINKIN, RUVBL1, RUVBL2, and α-tubulin were identified using a human cell line, and were also required for gonad cell adhesion in a nematode worm, these interactions may have a conserved molecular function in Metazoa including human.”